# Farmers' adoption of soil and water conservation practices: The case of Lege-Lafto Watershed, Dessie Zuria District, South Wollo, Ethiopia

Gizachew Shewaye Yifru[1], Birhan Asmame Miheretu[2]*

1 Agriculture Department, South Wollo Zone, Dessie, Ethiopia, 2 Department of Geography and Environmental Studies, Wollo University, Dessie, Ethiopia

☯ These authors contributed equally to this work.
* birhan1050@yahoo.com

## Abstract

In Ethiopia, soil degradation is one of the major causes of low and declining agricultural productivity. As a result of this challenge, the country has been battling to adopt conservation practices. The main objective of this study was to assess farmers' adoption decisions of soil and water conservation (SWC) practices. For the survey, 304 farmers were selected from farming communities in Lege-Lafto Watershed, South Wollo,Ethiopia. Information were gathered using a household survey, and through focus group discussions, key informant interviews, and field observation. A binary logistic regression model and descriptive statistics were used to analyze the data. The results indicated that about 64% of the farmers adopted soil and water conservation practices in the study area. The findings depicted that soil bund, stone bund, stone-faced soil bund, loose stone and brush-wood check dams, hillside terrace, and bund stabilized with vegetation are practiced in the watershed. The analysis result revealed adoption of soil and water conservation practices is significantly and positively influenced by the perception of farmers on erosion problems and SWC practices, family labour, educational level, and membership in local institutions. However, distance from residence to the nearest market and farmland, off-farm activities, and the ratio of cultivable land to family size influenced the adoption of SWC practices negatively. Therefore, improving farmers' educational status, and strengthening local institutions are vital for sustainable land management practices in the country.

## Introduction

Agriculture is the mainstay of Ethiopia's economy where its production is highly dependent on natural resources [1]. Agricultural land is a scarce resource in the Ethiopian highlands while it is the fundamental basis of rural livelihoods [2]. Due to drought and the growing decline in the quality and quantity of natural resources, which are the main basis of subsistence agriculture, the living conditions of the rural poor in the Ethiopian highlands have deteriorated

**Data Availability Statement:** Data is availability within the paper and Supporting Information. We attached minimal data set within the Supporting Information.

**Funding:** The author(s) received no specific funding for this work.

**Competing interests:** The authors have declared that no competing interests exist.

[3]. Land degradation in rural Ethiopia is seen as one of the key factors for low productivity and food insecurity [4–6].

Soil degradation is one of the most serious environmental problems in Ethiopia particularly in the highlands [4,7]. In the Ethiopian highlands, significant deforestation of natural forests, high population pressure, unsustainable farming practices, and conversion of marginal land to agriculture and grazing land have resulted in soil degradation [7–9]. The Ethiopian highlands have been experiencing declining soil fertility and severe soil erosion due to population pressure and the encroachment of the intensive farming system on steep and fragile lands [1,10].

A national assessment of soil erosion in the 1980s found that roughly half of the land area of the highlands (roughly 27 million hectares) was moderately eroded; 14 million hectares were severely eroded. Unfortunately, the "point of no return" has crossed more than 2 million hectares of farmland in the sense that it was unlikely that economic crop production could be maintained in the future [11]. Plot-level soil loss assessment also shows the severity of the problem which indicated that average soil loss rates on croplands have been estimated at 42 t ha$^{-1}$ yr$^{-1}$ but may also reach up to 300 t ha$^{-1}$ yr$^{-1}$ in individual fields [7]. Recently, the mean rate of soil loss by water erosion in Ethiopia was estimated as 16.5 t ha−1 yr−1, with an annual gross soil loss of ca. $1.9 \times 109$ t, of which the net soil loss was estimated as ca. $410 \times 106$ t (22% of the gross soil loss) [12]. Such losses may lead to irreversible changes in soil productivity that have a direct effect on Ethiopia's food security situation, with farmers unable to tolerate further deterioration in soil productivity [13] and as a result of this, agricultural production in the Northern highlands of Ethiopia is low [14]. Thus, Ethiopia's highland agricultural system is deteriorating resulting from soil erosion and nutrient depletion [15].

To mitigate the depletion of soil and its long-term impact on its productivity, it needs to be managed by appropriate SWC measures [16]. Soil conservation is critically required in these areas [10]. Based on this reality, the government of Ethiopia and non-governmental organizations have invested much in SWC practices since the mid- 1970s to reduce soil erosion, to improve agricultural productivity and food security [17–19]. The largest SWC activities were implemented during the 1970s and 1980s by mobilizing farmers mainly through food-for-work programs, though the result of the effort is minimal as the practices were not adopted by farmers [18,20,21]. In between the years 1995 and 2009, soil and water conservation practice was considered as a part of the agricultural extension package, therefore, implemented through the classical top-down approach.

Moreover, the government of Ethiopia instituted a national physical SWC construction campaign since 2011 that has been running for two months (January and February) every year in the high potential as well as low potential areas. The campaign is aimed at mobilizing the community to construct the necessary structures following watershed conservation principles. This approach is intended to change the attitudes of the farmers and ensure that the SWC structures are sustainable and effective [17]. Notwithstanding substantial efforts to establish and encourage different types of SWC practices, land users have not been generally adopted and used on a sustained basis for various reasons [1,8,18,21–23]. The problem might be explained by the fact that the adoption of SWC practices is influenced by demographic, socio-economic, institutional, and biophysical factors that are unique and complex in the area [1,22,24]. These studies on the decision to invest in SWC practices are not complete. For instance, [1] used the stone terrace to determine the farmers' adoption of SWC practices. However, considering a single conservation technology to determine the adoption of SWC practices is not complete as every conservation technology is not applied everywhere [16,25]. The adoption and diffusion of sustainable agricultural practices have become an important issue in the development-policy agenda for sub-Saharan Africa, especially as a way to tackle land degradation, low agricultural productivity, and poverty [26,27].

Moreover, South Wollo is characterized by a shortage of farmland, poverty, and recurrent drought [28]. In South Wollo, soil and water conservation measures continue to be adopted at lower rates than expected despite the considerable investments in reducing land degradation [29]. This constitutes one of the key research agendas in the country. Furthermore, there was no study regarding the adoption of SWC measures in the Lege-Lafto watershed, which is characterized by frequent erosion hazards. Any intervention for SWC and sustainable land use ought to begin with an empirical and local-specific understanding of the multiple factors affecting conservation decisions of farmers [18]. Cognizant of this, the study aimed at identifying the determinant factors influencing farmers' adoption decisions of soil and water conservation practices in Lege-Lafto Watershed, Dessie Zuria district, South Wollo, Ethiopia.

Hence, this article tried to answer what are the factors that determine the adoption of SWC measures in the study area? Thus, scientific analysis of SWC measures is of paramount importance to adopt SWC practices and would help for designing area-specific land management strategies. Designing land management methods based on demographic, socioeconomic, institutional, and biophysical characteristics could help to alter current land management practices or develop acceptable land management choices for a given location. This is especially crucial true in the study area, which included a wide range of demographic characteristics, land management practices, and soil fertility. Moreover, information on this subject is critical for the country's sustainable land management initiatives, which are a top priority for overall agricultural development programs.

## Materials and methods

### Description of the study area

This study was conducted at Lege-Lafto watershed of Dessie Zuria district which is located in the South Wollo zone, Ethiopia. The watershed is located 37 km far from Dessie town to the southwest and about 438 km northeast of Addis Ababa. Geographically, it lies between $10°56'$ 0" to $11°$ $9'$ 0" N Latitude and $39^0$ 25'20" to $39^0$ 33" 20' E Longitude (Fig 1). It is part of the headstreams of the Abay basin located on edge of the western escarpment of the rift valley and in the North-Eastern parts of central high lands of Ethiopia. The total area of the watershed is 24286 hectares. The slope of the watershed is flat to almost flat (1.0%), 5.6% is gently sloping, 13.6% is sloping, 33.0% is moderately steep sloping, 26.2% is steeply sloping and 20.6% is extremely steep sloping. The steep slope and rugged terrain of the watershed are natural hazards for soil erosion.

Based on 17 years (2002–2018) rainfall data obtained from Guguftu meteorological station ($10^0$ 54' 49" N, $39^0$ 29' 47" E), he mean annual rainfall of the area is 1365.7 mm with a maximum of 2074.4mm in 2018 and a minimum of 902.4mm in 2007. The mean annual temperature of the study area is 9˚c. The warmest month is June (10.4˚c) whereas the coldest month is January (7.4˚c) [30].

The type of farming system is mixed cereal-livestock. Barley *(Hordeum vulgare)* is the common type of crop in the upper part, while wheat (*Triticum*), teff (*Eragrostic tef*), maize (*Zea mays*), and sorghum (*Sorghum bicolor*) become widespread down to the lower. Legumes such as horse beans (*Vicia faba*), field peas (*Pisum sativum*), chickpeas (*Cisor arietinum*), and lentils (*Lens culinaris*) could be produced. Few farmers also engaged in producing potato (*Solanum tuberosum*), gomen (*Brassica carinata*), and onion (*Allium cepa*) from irrigated land. Livestock is intimately incorporated into the farming system. It supports crop production mainly as draught power in addition to the source of milk, meat, and cash income. Farmers generate income from off-farm activities. Selling of fire/pole woods, petty trade, and labour hire-out are sources of income for farmers.

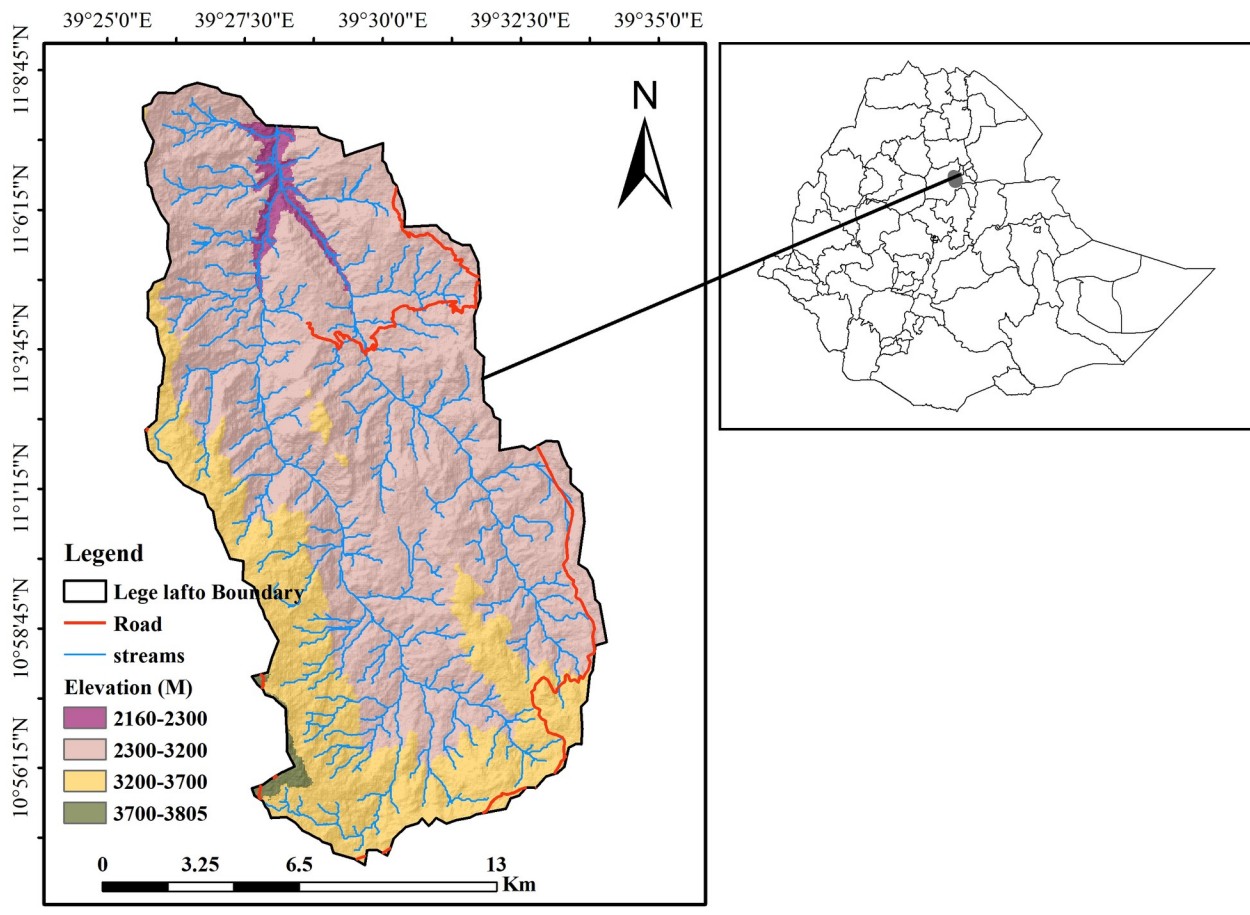

**Fig 1. Location of Lege-lafto watershed, Dessie Zuria District, South Wollo, Ethiopia.**

## Methodology

A concurrent mixed research design was employed in this study as qualitative and quantitative research methods were used side-by-side assuming both methods have equal importance.

### Sampling technique and selection procedure

Both purposive and systematic random sampling techniques were used to select the study area and sample households, respectively. First, Dessie Zuria district was purposely selected based on the discussion held with south wollo zone natural resource experts as this district is realized by its severe soil erosion-induced degradation problem, topographic variation, its accessibility, and long history in the implementation of different SWC measures. In the second stage, the study watershed was selected in the same manner based on the discussion held with Dessie zuria district natural resource experts. In the third stage, three *kebeles* were selected from the upper, middle, and lower sections of the watershed to represent the entire area. *Tebasite*, *Gelisha*, and *Asgedo kebeles* were systematically selected from the upper, middle, and lower sections of the watershed, respectively.

Households of selected *kebeles* are considered as the survey population and the number of sample households was decided by using the Yamane formula [31]. From the total 3251 farm household heads of *kebeles*, 357 respondents were selected by using a systematic random sampling technique. Finally, proportional numbers of sample respondents were selected from each

*kebeles*. Sampling was takenfrom lists obtained in the development agent offices. Fifty-three questionnaires were dropped out from analysis because of incompleteness and inconsistent responses. Hence, only 304 questionnaires were used for the analysis.

## Data source and data collection techniques

Questionnaires, in-depth interview, focus group discussion, and field observation was used as the main primary data collection techniques. Required data on demographic, socioeconomic, institutional, and biophysical factors of households were collected through a household survey questionnaire. Household-level and plot-level data were collected through open and close-ended questions survey questionnaires. The questionnaire included both closed and open-ended questions. The questionnaire enabled to collect data from a representative sample of household farmers. A structured questionnaire was designed for the interview also pretested with ten (10) respondents before the actual survey. The designed questionnaire covered a wide range of questions, which were intended to assess the adoption of SWC practices in the study area.

Besides household questionnaires, a focus group discussion (FGD) and key informants (KIs) interview were conducted to enrich information gathered through questionnaire survey and to capture information overlooked in the questionnaire. These tools were used to systematically generate information on the adoption of SWC practices they had observed and existing SWC measures. A total of three FGDs, one from each of the upper, middle, and lower watershed were conducted. Eight discussants including local elders, local opinion makers, and agricultural extension workers participated in each of the FGDs. These were individuals who have access to information on SWC measures and have been staying in the watershed for more than 25 years. A focus group checklist was administered in the local language. This checklist, comprised of open-ended questions around a set of SWC measures, was used to structure the discussions of participants' perspectives on how they implement different SWC measures in their land, and how this reduces soil erosion. Detailed notes were taken throughout the discussions. Besides, a total of 15 individual interviews with local community members were held to collect additional information and understand how their surroundings evolved, and learn more from their observations on SWC practices. We have chosen these key informants considering that they have adequate knowledge about the area. The informants are also thought that they can memorize the area's historical SWC practices. Field observations focusing on biophysical features, erosion indicators, farming practices, and conservation practices were carried out. In addition, secondary data were gathered from the South Wollo agriculture department.

## Methods of data analysis

Data gathered from sample households' were analyzed using descriptive statistics and a binary logistic regression model. Qualitative data gathered using focus group discussions, field observation, and key informant interviews were analyzed qualitatively by simple narrations.

Farmers' perceptions of erosion and SWC practices are analyzed by descriptive statistics. The result of the analysis is presented as frequency counts and percentage tables. An independent samples T-test is used to compare the mean difference between adopters and non-adopters of the practices. Furthermore, the chi-square test is used to check if there is an association between categorical variables and the adoption of SWC practices.

**Binary logistic regression model.** The logistic regression model is used to explore farmers' perception, demographic, economic, institutional, and physical factors influencing the adoption of SWC practices. A regression model, and its binary outcomes, help the researcher to explore how each explanatory variable affects the probability of the occurrence of events

[32]. This model helps to explore the degree and direction of the relationship between dependent and independent variables in the adoption of improved soil conservation technology at the household level [32]. The logistic regression model is an appropriate statistical tool to determine the influence of independent variables on dependent variables when the dependent variable has only two groups (dichotomous), e.g., adopters and -non-adopters, and the explanatory variables are continuous, categorical and dummy [4,17,29,32–34]. It enabled us to determine the impact of multiple independent variables on the dependent variable [33]. In the logistic model, the coefficients are compared with the probability of an event occurring or not occurring and bounded between 0 and 1. The dependent variable becomes the natural logarithm of odds when a positive choice is made. The result of binary helped to investigate the degree and direction of the relationship between the dependent and independent variables in the adoption of the practices at the household level. If the estimated values of the variables are positive and significant, farmers with higher values for the variables are more likely to adopt the soil and water conservation practices.

The model is specified as [35]:

$$\ln \left( P_x / (1 - P_x) \right) = \beta_0 + \beta_1 X_{1i} + \beta_2 X_{2i} + \cdots \beta_k X_{ki}$$

Where, the subscript i is the $i^{th}$ observation in the sample, $P_x$ is the probability of an event occurring for an observed set of variables $X_i$ i.e. the probability that farmer adopts the practices, and $(1 - P_x)$ is the probability of non-adoption. $\beta_0$ is the intercept term and $\beta_1, \beta_2 \ldots \ldots \beta_k$, are the coefficients of the explanatory variables $X_1, X_2 \ldots \ldots X_k$.

Therefore, the above model is used in this part to identify factors influencing the adoption of SWC practices in the study area. Before running the regression model multicollinearity among the explanatory variables is checked.

### Definition of variables and working hypothesis

**Dependent variable.** SWC technologies that are introduced in the area include soil, stone, and stone-faced soil bunds, loose stone and brushwood check dams, hillside terraces, and stabilized bunds with multi-purpose species. In this study, a farmer that constructs and maintains at least one of the introduced SWC technology either as recommended or with some modification is defined as the adopters. Therefore, a value of "1" is assigned to all households that constructed and maintained the technologies on his/her farmland (the "adopters"), and "0" is assigned to households who used indigenous and/or not- maintained the introduced technologies (the "non-adopters").

**Selection of explanatory variables and expected impact on adoption.** The adoption of SWC practices is a complex process that could be influenced by attitudinal, social, economical, institutional, and physical factors. Previous studies indicated that factors that affect adoption decision of farm household related to demographic, socioeconomic, institutional, and plot characteristics [1,3–5,15,17,22,29,34,36]. A range of independent variables that influence the adoption decisions of SWC measures by a farmer was identified based on a review of related literature. Accordingly, the descriptions of independent variables were indicated in Table 1.

The quantitative data was first entered into computer-based Statistical Package for Social Sciences (SPSS) software (version20) and the statistical analysis was done using this software.

### Ethics approval and consent to participate

Ethical clearance was obtained from the ethical review committee of Wollo University College of Medicine and Health Sciences. Permission to conduct the study was obtained from the South Wollo Zone Agriculture office and in turn, permission was secured from the Dessie

**Table 1. Definition of explanatory variables used in the model.**

| Variable | Descriptions |
|---|---|
| ADOPTER | Introduced SWC practices adopted; 1 if a farmer constructed and maintained the introduced SWC practices on his/her farmland, 0 otherwise |
| **Attitudinal factors** | |
| PRCPTERO | Perception of soil erosion as a problem; 1 if a farmer had perceived erosion as a problem, 0 otherwise |
| PRPTISWC | whether a farmer anticipates the introduced SWC practices are effective in mitigating erosion problem; 1 if a farmer anticipates soil erosion problem is resolved due to technologies and 0 otherwise |
| **Demographic factors** | |
| AGEHH | Age of the farm household head |
| FAMLSIZ | Number of people in the household |
| EDUCTION | Literacy of the household head; 1if literate and 0 otherwise |
| GNDER | Gender of the household head; 1if male and 0 otherwise |
| DEPRATIO | The ratio of total family number to the working-age group |
| FMLABOUR | Number of family members that are at working–age |
| **Institutional factors** | |
| TENURE | If the farmer feels that the land belongs to him/her at least in his/her lifetime; 0 otherwise |
| MMBRSHIP | Membership in local organizations; 1if a farmer is a member and 0 otherwise |
| OWNRSHP | Whether farmland is owned by the farmer; 1 if a farmer owns and 0 otherwise |
| CREDIT | Whether a farmer needed credit and was able to get it; 1 if he/she accessed 0 otherwise |
| TRNGSWC | Whether training SWC practice received by the farmer;1 if a farmer got training and 0 otherwise |
| EXTENSIN | Extension contact: 1 if the farmer gets extension services on SWC practices, 0 Otherwise |
| MARKDST | Walking distance to the nearest market |
| **Economic factors** | |
| LNDRTIO | Ratio of cultivable land to family size (hectare per person) |
| OFFINCOM | Whether a farmer engaged in off-farm employment, 1 if a farmer has off-farm employment and 0 otherwise |
| TLU | Number of livestock's in Tropical livestock unit |
| FRMSIZE | The size of the farm, in hectares |
| **Physical factors** | |
| FLMSLOPE | The slope of the plot as perceived by a farmer; 1 if steep and 0 otherwise |
| FRMQLTY | Soil fertility status of the plot; 1 if poor, 0 otherwise |
| FRMLDIS | The average distance of a plot from the homestead, in minutes |

Zuria District office. Before the data collection, the purpose of the study was explained to the study participants, and assurance was given that their participation in the study was voluntary. Then, informed written consent was obtained from each study participant. The confidentiality of the study participants' responses was ensured by not disclosing any information to a third party.

# Results and discussions

## Types of SWC practices that were implemented in the study area

Most of the surveyed farmers (86.5%) believed that soil erosion could be controlled. In line with this result, previous studies [18,37,38] reported that the majority of the surveyed farmers

confirmed erosion could be controlled. During the focus group discussion and key informant interviews farmers have identified the types of SWC measures in the watershed. These are a mix of traditional and/ or modern SWC practices. The traditional SWC practices such as cultivating along the contour and traditional ditch have been employed by farmers. Contour ploughing is used independently or in combination with other conservation practices such as stone and soil bunds. It has been carried out by using the ox-drawn plough. Hence, it is considered as an element of normal farming practices and the practice needs no extra labor and time for construction and maintenance like terraces. The traditional ditch is used for removing excess water from cultivated plots to waterways in a non-erosive way and was implemented except on extremely sloping plots in every cropping season. Hence, it is considered as a production practice designed to minimize the waterlogging problem rather than a conservation practice.

Improved SWC practices were introduced since the mid- 1970s in the study area. The SWC practices have been implemented by the farmers and these were mostly physical conservation technologies. These technologies include loose stone check dam, soil bund (**Fig 2**), stone bund (**Fig 3**), stone-faced-soil bund (**Fig 4**), hillside terrace (**Fig 5**), and brush-wood check-dam (**Fig 6**). Biological SWC measures including bund stabilization using grasses and/or legume shrubs are also implemented by farmers (**Fig 7**).

The study showed that 64.5 percent of the farmers have constructed and maintained SWC technologies. As indicated in **Fig 8**, 23.5, 35.2, 27.6, 6.6, 5.1, 1.5, and 0.5 percent of the farmers, respectively adopted soil bund, stone bund, stone-faced soil bund, loose stone check dam, bund stabilization using grasses and/or legume shrubs, hillside terraces and brushwood check-dams.

Participants of the focus group discussion mentioned that 'the *constructions of terracing and soil and stone bunds are what we learned from our fathers, just like we inherited lands from them. They added that soils that exist on each farmer farmland should be protected and we should seriously think about how to maintain the soils on that specific plot because if we are unable to fully stop erosion, the small amount that is being eroded every year has considerably reduced the thickness of soils that we have on our plots at present'.*

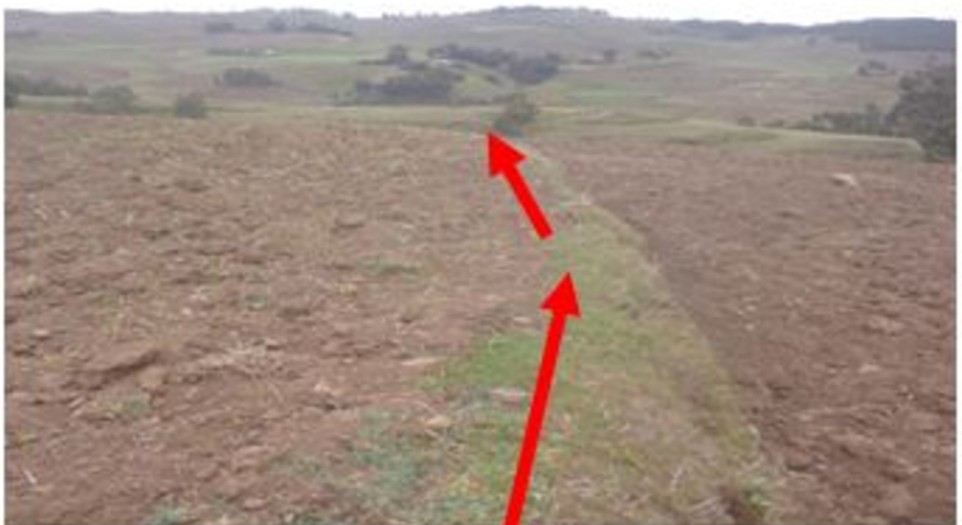

**Fig 2. Up-graded soil bund in study area.**

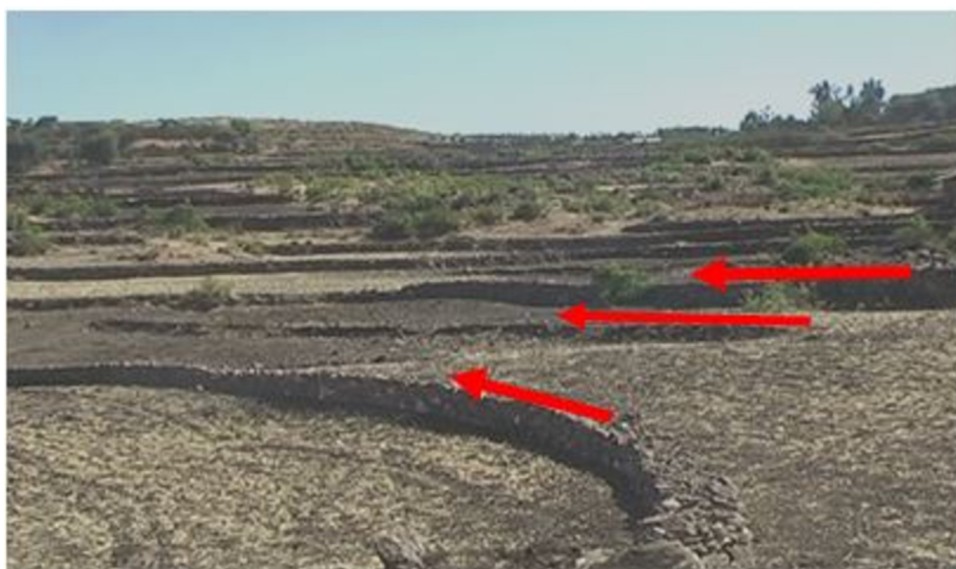

**Fig 3. Stone bund in the study area.**

These conservation activities are performed by mobilizing the masses for two months after the harvesting season. It has been promoted and implemented through community mass mobilization in the study area, as part of Ethiopia's Growth and Transformation Plans (GTP I and II) [39,40]. GTPs are a national development framework for five year periods: GTP I (2010/11 to 2014/15) was directed towards achieving the Millennium Development Goals by 2015 [22], and GTP II (2015/16 to 2019/20) was directed towards achieving the country's vision of becoming a middle-income country by 2025 [41]. The main sustainable land management (SLM) practices implemented through (community) mass mobilization include physical measures, such as stone/soil bunds, terraces, and check-dams, as well as biological

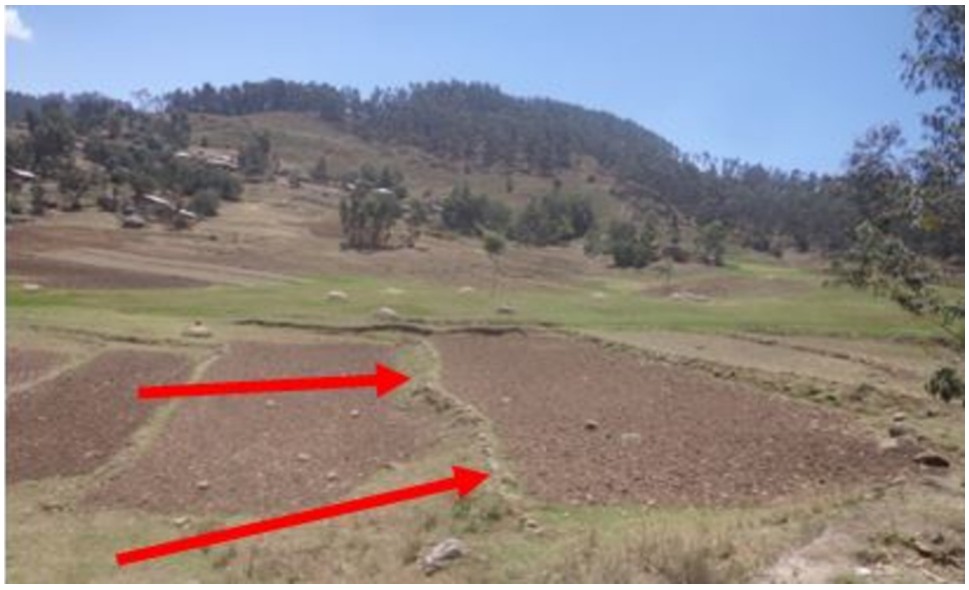

**Fig 4. Stone-faced soil bund.**

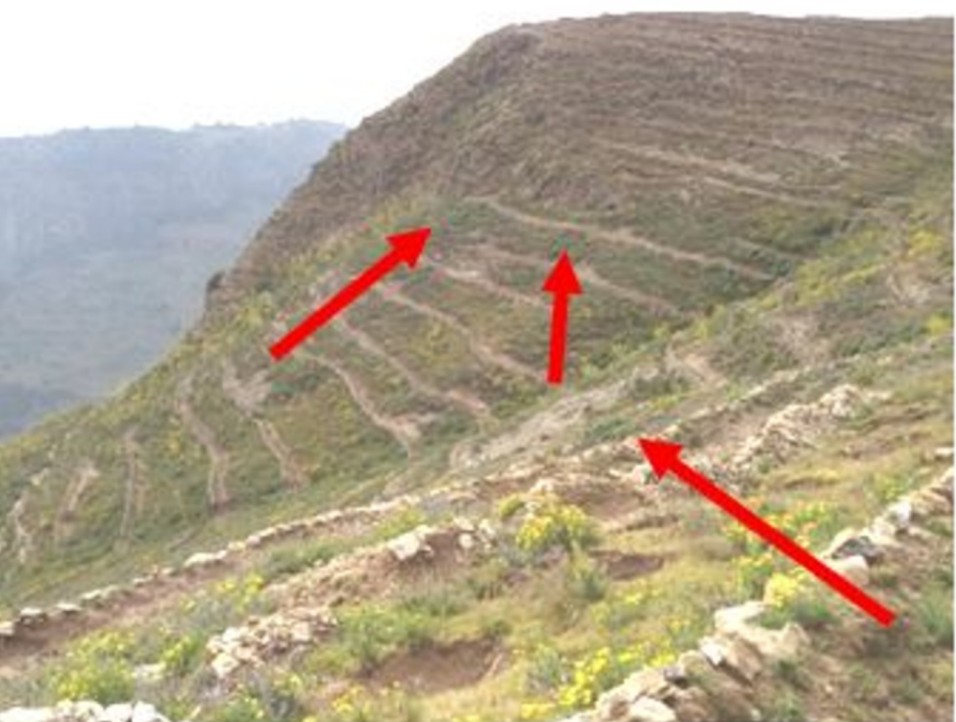

**Fig 5. Hillside terraces.**

measures, such as tree planting and area enclosures [40]. The farming communities in the rural areas were mobilized to implement both physical and biological SWC measures on individual and communal lands. These activities were intended to alleviate and reverse environmental problems in general, and soil erosion, land degradation, and deforestation in particular. This practice has in turn increased agricultural productivity and vegetation cover,

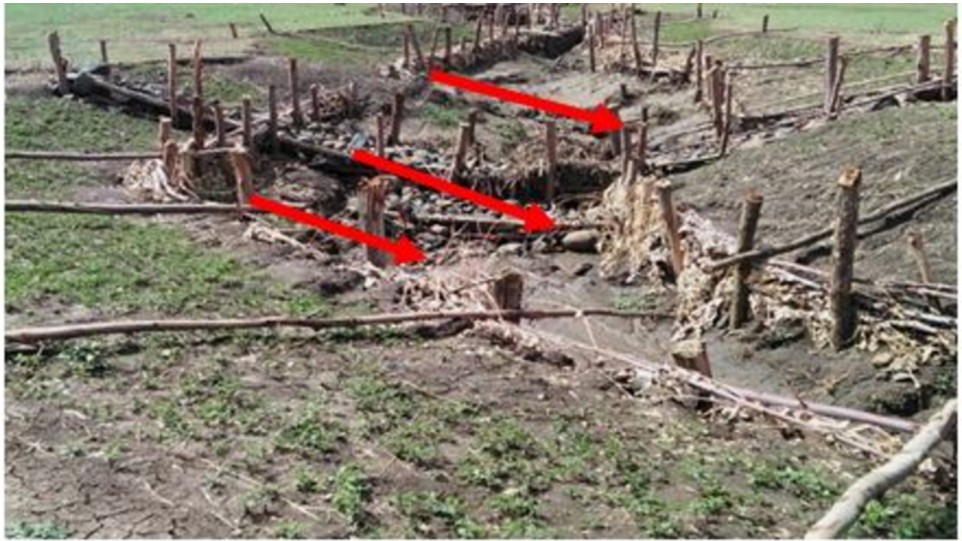

**Fig 6. Brush wood check dam.**

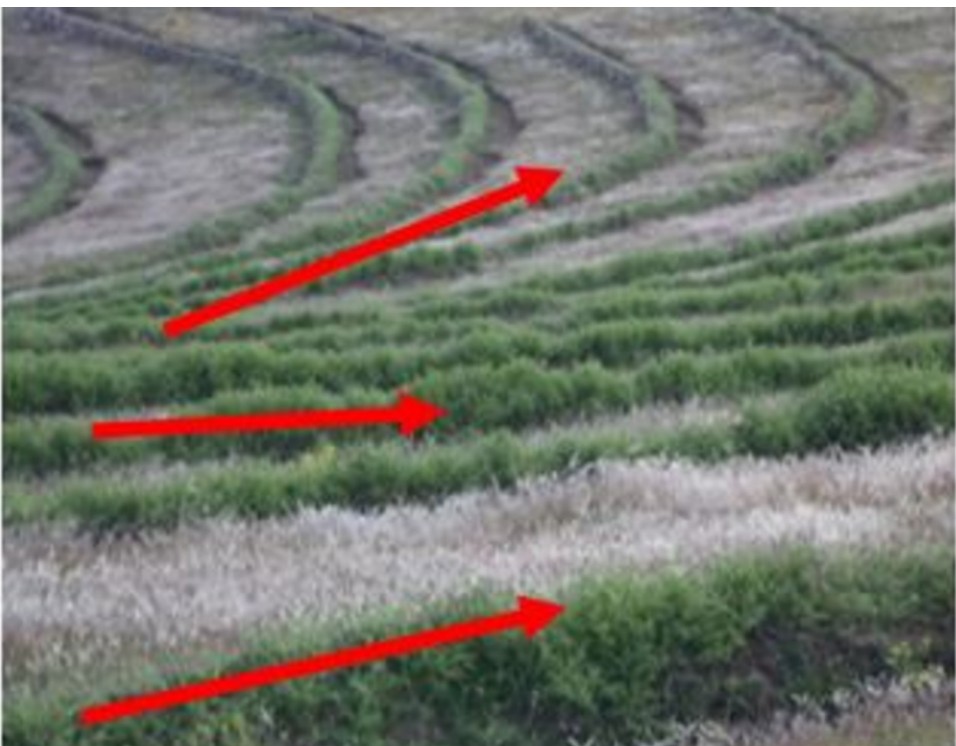

**Fig 7. Stabilized bund with *Desho grass*.**

helped to maintain biodiversity, and meet the growing fuelwood and construction wood demand of local communities.

"About 35 years ago, soil erosion was the most important concern in our area," said an old resident from the study area. Recognizing the situation, Ethiopia's then government introduced the concept of SWC measures, even though we had already done so. We did not reject the government's concept because we were searching for a solution, therefore we built the requisite SWC structures. Finally, we are able to maintain the soil in our cultivated area and begin producing good crops. We now know that we have the solution in our hands and we confirmed that different SWC technologies implemented by the mass mobilization campaign not

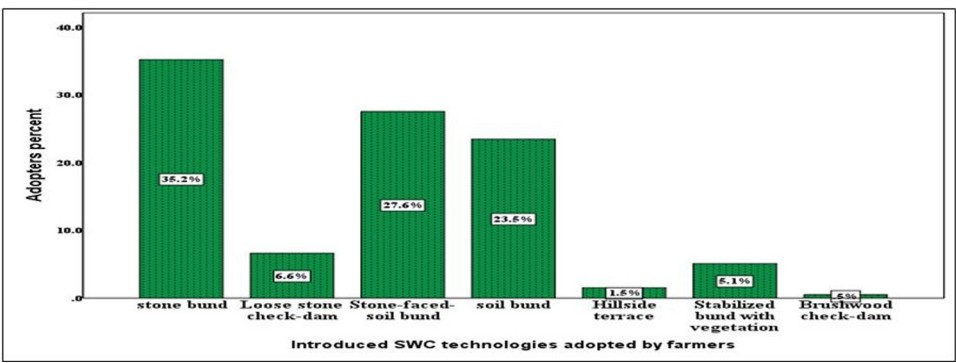

**Fig 8. Soil and water conservation practices in Lege-lafto watershed.**

only in decreasing erosion problems on the farmlands, but also in enhancing soil productivity and crop yields.

This implies that farmers were already familiar and/or involved, in mass mobilization SWC activities. This could be one of the factors that affect the adoption of SWC measures in the study area. According to [42], the current mass mobilization approach in Ethiopia can significantly enhance its impact on SLM and will be more sustainable. The adapted approach enhanced the awareness of farmers, created intrinsic motivation, fostered implementation of SLM practices in the field, and built responsibility in controlling erosion and reducing drought.

## Farmers' perception of soil and water conservation practices

Over 97% of the respondents perceived that SWC practices could improve agricultural production, soil loss reduction (91.8%), control flood (96.7%), improve soil fertility (88.5%), create a better farming plot (61.8%), and become the source of fuelwood and forage (77.0%). This finding is in line with [37] who reported that 93.5% of farmers perceived the positive role of the practices for reducing soil erosion problems.

During the group discussion and key informant interview held with farmers, the effectiveness of SWC practices was emphasized as they observed better growth and development of crops mainly along conservation technologies where fertile sediments were trapped. They also evaluated the amount of sediment trapped by soil and water conservation practices were very high which would be taken away out of the field if those conservation technologies were not constructed. Moreover, the majority of the farmers (93.4%) expressed interest to use the SWC practices on their land. In order to know the perception of farmer's about the importance of soil and water conservation, the researchers made an in-depth interview with a farmer X, a household head who have been living in the study area for more than 30 years. *Accordingly, 'he said that' he has 3 family members and the household has less than 1 hectare of land. He said that there is changes in the land condition since the SWC technologies were implemented in the study area. He witnessed improvement in vegetative cover of the watershed, reduced rate of soil erosion and improved growth of crops along the SWC structures.*

However, the major constraint cited by farmers to construct and maintain SWC practices were the technologies requiring intensive labor. Similarly, studies by [18,43] reported that 92 and 88 percent of farmers were respectively, constrained by the technology being labor-intensive for construction.

## Adoption status of soil and water conservation practices

Adopters and non-adopters differed in eight of the nine hypothesized continuous variables (Table 2). The average age of adopters and non-adopters was about 45 and 46 years, respectively. The mean walking distance measured in minutes between farmers' homesteads and the nearest market was 52.4 and 58.6 minutes for adopter and non-adopters, respectively. It showed that adopters had lower walking distance than non-adopters and the difference between the two groups (adopters and non-adopters) was statistically significant. This implies the adoption of SWC practices is enhanced as the walking distance of farmers' residences and nearest markets decreases.

In terms of household size, adopters were shown to contain slightly more household size than non -adopters. The results revealed that adopters, on average, had 4.86 persons compared to the 4.52 persons of non-adopters and the difference among them was statistically significant. This showed that the adoption of SWC practices increased as the size of a household increased. Adopters had slightly more farmland size than non-adopters. Descriptive statistics showed

**Table 2. Continuous variables differing adopters from non-adopters of SWC technologies in lege-lafto watershed.**

| Variables | Adoption category of SWC practices | | | | t-value |
| | Adopters | | Non-adopters | | |
| | Mean | Standard Deviation | Mean | Standard Deviation | |
|---|---|---|---|---|---|
| Age of the household (in years) | 44.6 | 9.1 | 45.6 | 9.8 | -0.84(ns) |
| walking distance to nearest market (in minutes) | 52.4 | 17.12 | 58.6 | 17.36 | 3.03** |
| household size (in numbers) | 4.86 | 1.29 | 4.52 | 1.41 | 2.12** |
| Farm land size (in hectares) | 0.86 | 0.42 | 0.78 | 0.46 | 1.53* |
| Cultivated land–to-man ratio | 0.14 | 0.90 | 0.15 | 0.12 | 1.35* |
| Livestock holding (in TLU) | 3.37 | 1.47 | 3.10 | 1.59 | 1.45* |
| Average walking distance of farm land from homesteads (in minutes) | 7.3 | 3.40 | 8.8 | 3.86 | 3.39*** |
| Dependency ratio | 1.87 | 0.751 | 2.79 | 1.35 | 6.66*** |
| Working-age with in a household (in number) | 2.91 | 1.096 | 1.83 | 0.791 | 9.84*** |

Note

***, ** and *indicates significant at 1, 5 and 10 probability level, respectively.

that adopters, on average, had 0.86 compared to 0.78 of non-adopters and the difference among the groups was also statistically significant. This implies as the size of farmland increases so does the adoption of SWC practices. The results also showed that adopters had a 0.14 land-to-man ratio while non-adopters had 0.15 and the difference among groups (adopter and non-adopter) was statistically significant. This verifies as the ratio of land-to-man ratio decreases the adoption of SWC practices increases. This result is in line with the previous study by [36] who reported that non-adopters had a higher land-to-man ratio (2.15) compared to the adopters (1.57), which is significantly influenced adoption decisions [36].

Moreover, as indicated in Table 2, adopters, on average, had 3.37 TLU compared to 3.10 of non-adopters and the difference was also statistically significant. This indicates as the farmer holds more livestock the adoption of SWC practices significantly enhanced. This result suggests that as the distance between farmers' homesteads and farmlands increased adoption of soil and water conservation practices become less. The dependency ratio was found to differ significantly between adopters and non-adopters. The results showed that as the number of dependent family members in a household increased, adoption SWC practices become less or vice versa. In addition, the findings indicated that as the number of working-age members in a household increases so does the adoption of SWC practices.

The results of the study indicated that 86.2% of adopter and 74.1% of non- adopter perceived a problem of soil erosion in their land. The chi-square test showed that perceiving a problem of soil erosion demonstrated a significant association with the adoption of SWC practices. This indicates the adoption of SWC practices increased when farmers perceived the problem of soil erosion on their farmland. The implication is that farmers who believe their farmlands are susceptible to soil erosion are more likely to use SWC measures than farmers who do not believe soil erosion is a concern. The findings also showed that non-adopters were more employed in off-farm activities (67.6%) than adopters (44.4%). This implies as farmers are employed in off-farm activities the adoption of SWCpractices becomes significantly smaller.

Both extension service and training on conservation practices enhanced the adoption of SWC practice. Regarding the source of farmland, 88.3% of adopters and 76.9% of non-adopters possess their farmland. The result of the chi-square test disclosed that land ownership is significantly associated with the adoption of SWC practices. This is in good agreement withthe

results of previous studies in Ethiopia that report the importance of land ownership in the adoption of SWC practices [5,15,47].

The results of the chi-square test in Table 3 also showed that membership in the local institution and credit access are significantly associated with the adoption of SWC practices.

**Table 3. Categorical variables distinguishing adopters from non-adopters of SWC practices in Lege-lafto watershed.**

| Variables | Adoption category of SWC practices | | $X^2$—test |
|---|---|---|---|
| | Adopters (%) | Non-adopters (%) | |
| Soil erosion is a problem on the plot | | | |
| Yes | 86.2 | 74.1 | 6.94*** |
| No | 13.8 | 25.9 | |
| Perception on the effectiveness of SWC practices | | | |
| Yes | 78.6 | 64.8 | 6.80*** |
| No | 21.4 | 35.2 | |
| Household head gender | | | |
| Male | 85.2 | 88.0 | 0.445 |
| Female | 14.8 | 12.0 | |
| Educational status of household head | | | |
| Illiterate | 44.9 | 57.4 | 4.36 |
| Literate | 55.1 | 42.6 | |
| Involvement in off-farm activities | | | |
| Yes | 44.4 | 67.6 | 15.04*** |
| No | 55.6 | 32.4 | |
| The farmer feels certain not to lose his/her farmlands | | | |
| Yes | 47.4 | 37.0 | 3.07 |
| No | 52.6 | 63.0 | |
| Source of farmland | | | |
| Owned | 88.3 | 76.9 | 6.82*** |
| Rented | 11.7 | 23.1 | |
| Get extension service on SWC practices | | | |
| Yes | 77.6 | 62.0 | 8.32*** |
| No | 22.4 | 38.0 | |
| Farmer attained training on SWC practices | | | |
| Yes | 45.4 | 26.9 | 10.1*** |
| No | 54.6 | 73.1 | |
| Credit access | | | |
| Yes | 46.4 | 30.6 | 7.26*** |
| No | 53.6 | 69.4 | |
| The household is a member of any local institution | | | 24.95*** |
| Yes | 80.1 | 52.8 | |
| No | 19.9 | 47.2 | |
| The perceived slope of the farmland | | | |
| Sloppy | 81.1 | 67.6 | 7.05*** |
| Flat | 18.9 | 32.4 | |
| Farmland quality | | | |
| Poor | 52.6 | 38.0 | 5.94 |
| Fertile | 47.4 | 62.0 | |

Note: *** indicates significant at less than 1 probability level.

## Determinants for the adoption of SWC measures

The logistic regression model was used to identify the factors that influenced the adoption behaviors of farmers on SWC practices. Before running the logistic regression analysis, the existence of multi-collinearity among the explanatory variables was checked. The result of col-linearity statistics showed that no variable resulted in a tolerance value of less than 0.1. More-over, the result of the variance inflation factor (VIF) is very small (much less than 10). Consequently, all variables are included for the final analysis as there is no multi-collinearity problem.

The estimated binary logistic regression model coefficients, Wald test, standard error, and significance levels are presented in Table 4. Regarding the performance of the model, the likeli-hood ratio test statistic exceeds the chi-square critical value (169.514) with 22 degrees of free-dom. The result was highly significant (P = 0.00) suggesting well fitness of the model. Similarly, the output of the Hosmer and Lemeshow (H-L) goodness-of-fit test indicated a non-significant probability (p = 0.102) value signifying the good fitness of the model. The compari-son among the correctly predicted observations with the constants included in the model also indicated that 82.9% of the observations were correctly predicted at 0.5 cut value. Correctly predicted adopters and non-adopters of the model are 90.8% and 68.5%, respectively. The pseudo $R^2$ was 0.587 also indicating the predictor variables in the model explained about 58.7% of the variation in the adoption of SWC practices. The results of the binary logistic

**Table 4. Binary logistic regression model estimates for adoption of SWC practices in Lege- lafto watershed.**

| Explanatory variables | $\beta$ | S.E. | Wald | P-value |
|---|---|---|---|---|
| PRPTISWC | 0.66 | 0.39 | 2.80 | 0.09* |
| EDUCTION | 0.97 | 0.38 | 6.66 | 0.01*** |
| TENURE | 0.20 | 0.36 | 0.29 | 0.59 |
| EXTENSIN | 0.60 | 0.39 | 2.34 | 0.13 |
| TRNGSWC | 0.57 | 0.40 | 1.97 | 0.16 |
| CRIDIT | 0.40 | 0.36 | 1.23 | 0.27 |
| OFFINCM | -1.06 | 0.36 | 8.54 | 0.00*** |
| MMBRSHIP | 1.66 | 0.48 | 11.94 | 0.00*** |
| PRCPTERO | 1.05 | 0.43 | 5.95 | 0.01*** |
| FRMQLTY | 0.36 | 0.34 | 1.12 | 0.29 |
| FLMSLOPE | 0.20 | 0.40 | 0.25 | 0.62 |
| GNDER | 0.03 | 0.55 | 0.00 | 0.95 |
| OWNRSHP | 0.48 | 0.49 | 0.95 | 0.33 |
| AGEHH | 0.02 | 0.02 | 0.69 | 0.41 |
| FRMLDIS | -0.10 | 0.05 | 4.33 | 0.04** |
| MARKDST | -0.03 | 0.01 | 9.73 | 0.00*** |
| FMLABOUR | 1.18 | 0.45 | 6.94 | 0.01*** |
| FAMLSIZ | -0.33 | 0.26 | 1.56 | 0.21 |
| LNDRTIO | -5.59 | 3.18 | 3.09 | 0.08* |
| DEPRATIO | -0.26 | 0.37 | 0.51 | 0.48 |
| FRMSIZE | 1.12 | 0.75 | 2.25 | 0.13 |
| TLU | -0.08 | 0.13 | 0.32 | 0.57 |
| CONSTANT | -2.05 | 1.99 | 1.05 | 0.30 |

Note

***, ** and * Significant at 0.01, 0.05 and 0.1 probability levels, respectively.

regression model analysis revealed that the adoption of SWC practices is influenced by several factors. The factors that positively and significantly influenced the adoption of SWC practice were perception on soil erosion problem, educational level and local institution membership, family labour while engagement in off-farm activities, walking distance between farmland and homestead, the distance between nearest market and home, and land- to- man ratio were found to be negatively and significantly influencing the adoption of SWC practices (Table 4).

The adoption of SWC practices increases with the perception of farmers about the positive role of SWC practices to mitigate soil erosion. This shows that perception of the effectiveness of SWC practices is a necessary condition to adopt the practices. This result is in line with the findings of [1,4]. who reported that farmers' perception of technology attributes influenced positively and significantly their adoption decision on conservation practices.

The result of binary logistic regression analysis depicted that the perception of soil erosion as a problem affects the use of SWC practices positively and significantly ($\beta$ = 1.05; p-value = 0.01) which is also confirmed by the Wald statistics (5.95). This indicates that farmers who perceived the problem better are expected to invest more in conservation practices. Similarly, [44] indicated that farmers who had already perceived their land to have soil erosion problems were more likely to adopt SWC measures than those who did not.

The educational level of farmers increases their ability to get and use the information and improves farmers' decisions to adopt SWC practices. The farmers' education status affects their use of soil and water conservation practices positively and significantly ($\beta$ = 0.97; p-value = 0.01) among the farmers of Lege- lafto watershed. This finding corroborates with the finding of previous studies [4,29,45,46] who stated that a better education level of household heads has a strong and positive relationship with farmers' adoption of SWC practices.

Being membership in a local organization (Development groups and watershed user association) assists a person to obtain information on improved farming practices. This factor is positively and significantly (($\beta$ = 1.66; p-value = 0.00) which is also confirmed by the Wald statistics (11.94)) influenced by the use of SWC practices. This means local organizations have an immense contribution to the adoption of conservation practices. This is consistent with the findings of [47].

The results showed that involvement in off-farm activities was negatively and significantly ($\beta$ = -1.06; p-value = 0.00; Wald statistics = 8.54) influencing the adoption of SWC practices in the study area. This implies that there is labor competition between off-farm activities and SWC practices that discourage farmers from engaging in the implementation and maintenance of conservation practices on their lands. This result is in line with the findings of other empirical studies [1,4,29,44]. However, it contradicts with the study from Nigeria [48] who reported that off-farm income exhibited a significant positive correlation with the adoption of land management technology because the income accruing from off-farm activities by farm household members who have migrated to other areas can help farmers to afford the cost of technology, thereby increasing the adoption overall.

The distance of farmland from the homestead negatively and significantly ($\beta$ = -0.10; p-value = 0.04) which is also confirmed by the Wald statistics (5.95) influenced farmers' adoption decisions. This implies that a unit increase in walking distance between farmland and residential area lowers significantly the use of conservation practices. Farmers want to invest more in the nearest farmlands as far away farmlands discouraged them to construct and maintain soil and water conservation practices. This result is in line with the previous studies [22,29,46,49] who reported that the longer the walking distance between farmland and residential area, the less likelihood of the adoption of SWC practices.

The land is the most important natural asset to the rural households studied. The per capita availability of land has been diminishing in many parts of the country due to the growing rural

population and loss of productive lands to degradation [47]. In seriously degraded and scarce farmland, vulnerability to starvation increases by decreasing land–to- man ratio. Therefore, it is assumed that the adoption of SWC practices increased with a decrease in cultivated land-to-man ratio (LNDRTIO). The findings indicated that land to man ratio had a statistically significant negative ($\beta$ = -5.59; p-value = 0.08) influence on farmers' adoption decisions which is also confirmed by the Wald statistics (3.09). This indicates that an increase in the land area per capita discouraged people to adopt conservation measures. One probable explanation is that as the arable land area grows, so do the expenses of implementing particular conservation measures, such as labor expenditures. This makes those practices less likely to be adopted. This is in line with the findings of [26,36], who found that the costs of implementing SWC measures grew with the size of arable land, posing a barrier to smallholder farmer adoption.

The results indicated that market distance was negatively and significantly ($\beta$ = -0.03; p-value = 0.00) affected the adoption of SWC practices. Similarly, a unit increases in walking distance between market and residential areas lower significantly the adoption of soil and water conservation (SWC) practices. This is consistent with an earlier study by [5].

Families are an important source of labour for farm operations and the construction of SWC practices. The findings showed that the family size (FMLABOUR) was positively and significantly ($\beta$ = 1.18; p-value = 0.01) affects the adoption of SWC practices. A unit increase in the number of working-age members increases significantly the use of SWC practices. This suggests that households that are endowed with much working-age family labor favor the adoption of SWC practices. This is in good agreement with the findings of previous studies by [5,50].

## Conclusions

The result of the study revealed that farmers perceive soil erosion as a problem and they believed that erosion could be controlled by adopting SWC practices. Different types of SWC practices were introduced in the watershed. Adopted SWC practices are soil bund, stone bund, stone-faced soil bund, loose stone check dam, brush wood check dam and hillside terrace, and bund stabilization with legumes and grass.

If other studies support the findings reported here, then the following policy implications emerge. A farmer's decision to adopt SWC practices was significantly associated with perception or increasingly severe erosion problems, educational level, and local institution membership. However, engagement in off-farm activities, walking distance between farmland and homestead, the distance between nearest market and homestead, and land- to- man ratio were found to be negatively and significantly influencing the adoption of SWC practices.

Therefore, to increase the likelihood of adoption, agricultural practitioners should organize and train farmers to effectively introduce conservation initiatives. Based on these results, we also suggest that efforts to resolve land degradation using SWC mechanisms should concentrate on the human and institutional capacity building of farmworkers. As a result, it's critical to design soil and water conservation methods that combine modern scientific understanding with indigenous technological knowledge in order to spread them and maintain their long-term viability.

## Supporting information

**S1 Data.**
(SAV)

## Acknowledgments

We are indebted to the farmers for sharing their experiences. We are also grateful to agricultural experts at Dessie Zuria district for sharing their ideas and for providing technical support. Last but not the least, we also thank the English language professional who is Asres Nigus (PhD) for the language editing of the manuscript.

## Author Contributions

**Conceptualization:** Gizachew Shewaye Yifru, Birhan Asmame Miheretu.

**Data curation:** Gizachew Shewaye Yifru, Birhan Asmame Miheretu.

**Formal analysis:** Gizachew Shewaye Yifru, Birhan Asmame Miheretu.

**Investigation:** Gizachew Shewaye Yifru, Birhan Asmame Miheretu.

**Methodology:** Gizachew Shewaye Yifru, Birhan Asmame Miheretu.

**Project administration:** Gizachew Shewaye Yifru, Birhan Asmame Miheretu.

**Resources:** Gizachew Shewaye Yifru, Birhan Asmame Miheretu.

**Software:** Gizachew Shewaye Yifru, Birhan Asmame Miheretu.

**Supervision:** Gizachew Shewaye Yifru, Birhan Asmame Miheretu.

**Validation:** Gizachew Shewaye Yifru, Birhan Asmame Miheretu.

**Visualization:** Gizachew Shewaye Yifru, Birhan Asmame Miheretu.

**Writing – original draft:** Gizachew Shewaye Yifru, Birhan Asmame Miheretu.

**Writing – review & editing:** Gizachew Shewaye Yifru, Birhan Asmame Miheretu.

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
