## [Decision Letter · Decision Letter 0]

16 Sep 2021

PONE-D-21-19431

Farmers’ adoption of soil and water conservation practices: The case of Lege-Lafto Watershed, Dessie Zuria District, South Wollo, Ethiopia

PLOS ONE

Dear Dr. Miheretu,

Thank you for submitting your manuscript to PLOS ONE. After careful consideration, we feel that it has merit but does not fully meet PLOS ONE’s publication criteria as it currently stands and needs considerable work. Therefore, we invite you to submit a revised version of the manuscript that addresses the points raised during the review process.

See comments which are provided in some detail by all three reviewers, these are to help guide your reanalysis and revised manuscript.

We look forward to receiving your revised manuscript.

Kind regards,

Sieglinde S. Snapp

Academic Editor

PLOS ONE

4. Thank you for stating the following in the Acknowledgments/Funding Section of your manuscript:

“This work was supported by South Wollo Agriculture office, Dessie , Ethiopia.”

“M.Gizachew received fund.

This work was supported by South Wollo Agriculture office, Dessie , Ethiopia.

No. The funder had no role in the study design, data collection, and analysis, decision to publish.”

7. We note that Figure 1 in your submission contain [map/satellite] images which may be copyrighted. All PLOS content is published under the Creative Commons Attribution License (CC BY 4.0), which means that the manuscript, images, and Supporting Information files will be freely available online, and any third party is permitted to access, download, copy, distribute, and use these materials in any way, even commercially, with proper attribution. For these reasons, we cannot publish previously copyrighted maps or satellite images created using proprietary data, such as Google software (Google Maps, Street View, and Earth). For more information, see our copyright guidelines: http://journals.plos.org/plosone/s/licenses-and-copyright.

8. Please include your tables as part of your main manuscript and remove the individual files. Please note that supplementary tables (should remain/ be uploaded) as separate ""supporting information"" files

Additional Editor Comments (if provided):

Please find three reviewers who have provided very detailed comments and advice. The manuscript requires considerable work, and the tables need to be included which somehow were not part of the manuscript uploaded. The updated model needs to be considered to reanalyze the data, that is the suggestion to model jointly the decisions to adopt different agricultural technologies, as is now common practice in the agricultural technology adoption literature (e.g., Kassie et al. 2013; Ward et al., 2018). This needs to be tried out in the revised manuscript.

The introduction requires much more careful description of the research gap, and the discussion also needs considerable revisions to clarify the key findings, as several reviewers noted.

Overall, this is an important study and we look forward to seeing a revised manuscript addressing these comments.

Reviewers' comments:

Reviewer's Responses to Questions

**Comments to the Author**

1. Is the manuscript technically sound, and do the data support the conclusions?

Reviewer #1: Partly

Reviewer #2: Yes

Reviewer #3: Partly

2. Has the statistical analysis been performed appropriately and rigorously? 

Reviewer #1: Yes

Reviewer #2: No

Reviewer #3: No

3. Have the authors made all data underlying the findings in their manuscript fully available?

Reviewer #1: No

Reviewer #2: No

Reviewer #3: No

4. Is the manuscript presented in an intelligible fashion and written in standard English?

Reviewer #1: No

Reviewer #2: Yes

Reviewer #3: No

5. Review Comments to the Author

Reviewer #1: Major issues:

1. The Introduction and Methodology sections are clear, but the Discussion of key results is very weak. Results need to be discussed in a way that brings out the ‘big picture’ implications of the key findings. Reference to literature from within and outside Ethiopia needs to be improved.

2. The Results and Discussion section has numerous grammatical errors. The authors are advised to seek editing assistance from a native English speaker.

Minor issues:

1. The document requires line numbers in order to facilitate easier review process

2. Avoid using words that already appear in the title as key words.

3. Introduction section

• Do not start sentences with ‘And’

• Change tone ha-1 yr-1 to t ha-1 yr-1

• Farmers are mobilised through food-for-work programs, not through food & work programs.

• Once the abbreviation for ‘Soil and Water Conservation’ has been defined in the Introduction, then use SWC throughout the rest of the document. There are instances where even both are used in the same sentence, avoid that.

4. Materials & Methods section

• Scientific names of crops should be accompanied by the naming authorities

• Clarify how the selection of each kebele in the different watershed positions was done. Was it random or systematic?

• Replace ‘plot of land’ by ‘land’

• In the text, be consistent with the refencing style, either numbers or author name(s) followed by year. Check the journal requirement.

• The different levels of the sub titles should be clear.

Results

• Tables are missing

• Avoid putting table titles between paragraphs

Reviewer #2: This paper examines the factors associated with adoption of sustainable soil and water conservation (SWC) practices in South Wollo, Ethiopia using primary survey data collected from 304 farm households. The data indicate that 64% of farmers adopted at least one of seven sustainable agricultural practices. The analysis finds that adoption is positively associated with family labor, education, and membership in local institutions. Factors found to correlate negatively with adoption are distance to market, engagement in off-farm activities, and the ratio of cultivable land to family size.

While the article focuses on an important policy issue – how to encourage adoption of sustainable agricultural practices – it is not immediately evident how the results contribute to scholarship or are of interest to policymakers and practitioners outside Ethiopia. The author might therefore consider submitting the paper to a regional agricultural economics journal after careful revision. Below I provide some suggestions for how to improve the paper.

The Introduction includes useful background on agriculture and natural resource degradation in Ethiopia but leaves out some essential components, including (a) a brief review of relevant literature and the research gaps that merit investigation, (b) a statement of the research questions to be addressed or hypotheses to be tested, and (c) how the current work contributes to scholarship. In terms of contributions, the authors mention that our knowledge of SWC technology adoption is incomplete, because many studies consider a single SWC technology rather than multiple technologies. This is inaccurate. There are many studies, including for Ethiopia, that consider adoption of multiple technologies. The other contribution mentioned by the authors is the focus on South Wollo. What is it about South Wollo that makes it particularly interesting as a case study? And what do we learn from this study that has relevance regionally or globally? Those are important questions to address when highlighting the study’s contributions.

The manuscript appears to be technically sound, and the data do support the conclusions. However, the analyses have not been performed rigorously. The regression model is a logit model of the adoption of at least one of the seven SWC practices under study. A better approach is to model jointly the decisions to adopt different agricultural technologies, as is now common practice in the agricultural technology adoption literature (e.g., Kassie et al. 2013; Ward et al., 2018). These studies show that by analyzing the complementarity and substitutability of agricultural practices the drivers of adoption and overall uptake are better understood. It is recommended that the authors review the study of Teklewold et al. (2013) who modeled the adoption of sustainable agricultural practices by Ethiopian maize farmers as a joint decision and incorporated the extent of such adoption (number of practices adopted). The authors should consider using this modeling approach.

Two other suggestions for the empirical analysis are as follows: First, more information should be provided on how some variables were measured in the survey, to give readers a sense of their accuracy. For example, was farm size measured by farmer report, tape/compass, or GPS device? How were the variables for distance to market and farmland measured? Second, the authors should include explanations for why the explanatory variables are included in the model, referring either to economic theory or previous empirical work.

Another possible way to increase the paper’s scholarly contribution would be to make better use of the qualitative data that were collected through focus group discussions (FGDs) and in-depth interviews. The authors claim the study is mixed methods, but they only mention a few qualitative results. As described by Greene et al. (1989), there are five broad purposes of MM research (development, triangulation, complementarity, initiation, and expansion). The authors could consider which of these purposes is/are of most relevance to their study and use that to guide how they make use of the qualitative data. In terms of complementarity, it would be useful to present both summaries and illustrative quotes from the qualitative data to add richness to the quantitative findings.

The Results section is written in a descriptive manner. This is acceptable, as long as a Discussion section is included, which is a requirement of PLOS One. The discussion is important for placing the results in context. In this section, the authors can return to the study hypotheses and discuss whether the findings are supportive of them. The discussion section should also refer to related studies and discuss how the findings of the present study agree or disagree with previous work and offer possible explanations. Finally, it is important to highlight the implications of the study findings for policy and practice.

As a final recommendation, I suggest the authors carefully edit the manuscript for brevity (there is repetition in some areas) and grammar.

References:

Greene, J. C., Caracelli, V. J., and Graham, W. F. 1989 “Toward a Conceptual Framework for mixed method evaluation designs.” Educational Evaluation and Policy Analysis 11: 255-274.

Kassie, M., Jaleta, M., Shiferaw, B., Mmbando, F., and Mekuria, M.. 2013. “Adoption of Interrelated Sustainable Agricultural Practices in Smallholder Systems: Evidence from Rural Tanzania.” Technological Forecasting and Social Change 80(3): 525–540.

Teklewold, H., Kassie, M., and Shiferaw, B.. 2013. “Adoption of Multiple Sustainable Agricultural Practices in Rural Ethiopia.” Journal of Agricultural Economics 64(3): 597–623.

Ward, P.S., Bellb, A.R., Droppelmann, K., and Bentond, T.G.. 2018. “Early Adoption of Conservation Agriculture Practices: Understanding Partial Compliance in Programs with Multiple Adoption Decisions.” Land Use Policy 70(1): 27–37.

Reviewer #3: Generally, such type of studies is timely because soil erosion is a significant environmental problem especially in countries like Ethiopia which depend on Agricultural activity which call for proper application of SWC activities. And identifying level of adoption and influencing factors are important. I thank you for sharing such local area-based study.

However, the manuscript has number of limitations to meet publication and proper contribution of knowledge:

- Poorly organized, and discussed (both abstract, background, result ….) and has a problem of flow of idea and coherence, some of them are redundant

- Soil erosion and the practice is dynamic but the data and citation is too old (1980 & 1990’s…..)

- Incorporate the contribution of this study to SDG’s and Millennium Development goals of the country .

- It couldn’t consider the contribution of 60 days Mass mobilization conducted for two consecutive months (January and February) in the adoption of the practices. It is also difficult to differentiate the adopters from non-adopters

- You wrote “there was no adoption of SWC measures study in the study area, and it has been frequently hit by erosion hazards” as a gap, however, there are number of studies conducted in simlar areas (south Wollo, Amhara region…) which have the same social and physical setup … and it is not unique from other areas in erosion hazard, if it is you should explain numerically

Thus, I didn’t see its originality.

In the Methodology part

- Justify the saying “even rains of moderate-intensity are enough to cause massive erosion” idea incorporate the annual soil loss rate and the erosivity and erodibility character of the RF and soil.

- The sampling lacks clarity . Example

o The way you select the study area is not clear (it needs to justify how much it is vulnerable for erosion? by how much than other districts in the zone? “Long history of conservation” by how much than other woredas?

- Usually, books and journals are used as literature not as data source. You stated that “secondary data were gathered from the department of Agriculture, books, and journals” . what type of data have you collected from books, journals and agriculture, where is the analysis of this data?

- In statistical analysis :

o Which type of T-test have used for what purpose and which variable have you analyzed ? I debt the fitness of it for this study

o How do you computed the dependency ratio, where is the result and data?

- The map of study area is not to the standard. It lacks aesthetic value. The coloring of topography is proper (use Elevation 1 or 2 color ramp from ArcMap). The inset maps are too many which are irrelevant ( one “Ethiopia “ ) is enough , the legend, the grids( conducted), titling, scale writing and placement, and it is deformed. Thus, improve and meet the standard of study area map

Result

- The model fitness and predictive capacity is very low (65% for non-adopters) why it is do you think the analysis result is reliable with this……………….; the multicollinearity?

- I didn’t find any table in the manuscript (Table 1; 2;3 and 4) but it is sited in the discussion text. This made difficult to see the correctness of statistical values (analysis outputs). Beside in the discussion part the result is discussed as “positively and significantly ..or …..” . without stating the value (p-value or wald statistics, and others…….) which shows how much the variable is significant than others. Generally, the statistical analysis result is not well interpreted and discussed with others study output.

- The discussion is poor and shallow it seems a report than a research study. Use recent studies and discuss it the similarity and difference of your study from them and why, and how?

6. PLOS authors have the option to publish the peer review history of their article (what does this mean?). If published, this will include your full peer review and any attached files.

Reviewer #1: No

Reviewer #2: No

Reviewer #3: No

---

## [Author Response · Author response to Decision Letter 0]

27 Nov 2021

Dear Sieglinde S. Snapp

Academic Editor

PLOS ONE

Subject: Ref.: No. PONE-D-21-19431

We are writing this with reference to the revised manuscript entitled Farmers’ adoption of soil and water conservation practices: The case of Lege-Lafto Watershed, Dessie Zuria District, South Wollo, Ethiopia, which has be submitted for publication in PLOS ONE. 

We would like to express our sincere gratitude to the reviewers for their valuable comments and suggestions. We have revised the manuscript by taking into account the comments and suggestions given by reviewers (Reviewers #1, #2and #3). We are now re-submitting the revised version for your kind reconsideration for publication. We sincerely hope that we have sufficiently addressed the suggested comments, and have prepared the manuscript in accordance with the journal’s style. The details of how the comments and suggestions as addressed point by point are given below. 

Looking forward to hear your positive response to publish our manuscript in your Journal, 

=====

Rebuttal letter

Response to the Journal Requirements

Response: We thank you for your key comments and we revised the manuscript based on POLS ONE manuscript preparation templates including file naming (Please see the revised version). 

Response: Thank you for this comment and we addressed the issue of language usage. As commented by the reviewers and academic editor, the manuscript is reviewed and edited by English language editor (professional). His name is Asres Nigus (PhD) who is working at Wollo University, Department of Teaching English As a Foreign Language, Dessie, Ethiopia. Hence, we believe that the language of the current version of the paper is significantly improved and consequently the quality of the paper too. (You can see in our acknowledgment section of revised manuscript with track changes document on page 24 from lines 591 to 592.

Ok

4. Thank you for stating the following in the Acknowledgments/Funding Section of your manuscript:

“This work was supported by South Wollo Agriculture office, Dessie , Ethiopia.”

“M.Gizachew received fund.

This work was supported by South Wollo Agriculture office, Dessie , Ethiopia.

No. The funder had no role in the study design, data collection, and analysis, decision to publish.”

We have removed funding information from the manuscript.

6. PLOS requires an ORCID iD for the corresponding author in Editorial Manager on papers submitted after December 6th, 2016. Please ensure that you have an ORCID iD and that it is validated in Editorial Manager. 

Response: I have validated. ORCID: https://orcid.org/0000-0003-1461-1411

7. We note that Figure 1 in your submission contain [map/satellite] images which may be copyrighted. All PLOS content is published under the Creative Commons Attribution License (CC BY 4.0), which means that the manuscript, images, and Supporting Information files will be freely available online, and any third party is permitted to access, download, copy, distribute, and use these materials in any way, even commercially, with proper attribution. For these reasons, we cannot publish previously copyrighted maps or satellite images created using proprietary data, such as Google software (Google Maps, Street View, and Earth). For more information, see our copyright guidelines: http://journals.plos.org/plosone/s/licenses-and-copyright.

Response: Figure 1 is removed from the manuscript

8. Please include your tables as part of your main manuscript and remove the individual files. Please note that supplementary tables (should remain/ be uploaded) as separate ""supporting information"" files

Response: The tables are included in the main manuscript.

Academic Editor and Reviewers’ comments and our responses 

 Additional Editor Comments (if provided):

Dear Academic Editor. Thank you for your summarized main concerns and suggestions that need to be addressed, and we updated the manuscript accordingly. 

Please find three reviewers who have provided very detailed comments and advice. 

The manuscript requires considerable work

Response: We have modified the manuscript

and the tables need to be included which somehow were not part of the manuscript uploaded.

Response: The tables are included in the main text of the manuscript

 The updated model needs to be considered to reanalyze the data, that is the suggestion to model jointly the decisions to adopt different agricultural technologies, as is now common practice in the agricultural technology adoption literature (e.g., Kassie et al. 2013; Ward et al., 2018). This needs to be tried out in the revised manuscript.

Response: We are grateful to the reviewers for his/her positive assessment of the methodology of the manuscript. We have given detail explanations for the model. (Please see revised manuscript with track changes document On page 8 line number 227-235)

The introduction requires much more careful description of the research gap, and the discussion also needs considerable revisions to clarify the key findings, as several reviewers noted.

Overall, this is an important study and we look forward to seeing a revised manuscript addressing these comments.

Response: Thank you for the positive remark on our paper and we really appreciate your scientific judgment. We have carefully addresses all the comments given by reviewers. (Please see revised manuscript with track changes document)

Reviewer's Responses to Questions

Comments to the Author

1. Is the manuscript technically sound, and do the data support the conclusions?

Reviewer #1: Partly

Reviewer #2: Yes

Reviewer #3: Partly

Response: Thank you very much for recognizing our study importance. We have modified the manuscript in accordance with the reviewers comment.

2. Has the statistical analysis been performed appropriately and rigorously?

Reviewer #1: Yes

Reviewer #2: No

Reviewer #3: No

Response: Thank you for the critical observation of statistical analysis. We have improved the statistical analysis procedures and interpretations in the revised manuscript based on the reviewers comment.

3. Have the authors made all data underlying the findings in their manuscript fully available?

Reviewer #1: No

Reviewer #2: No

Reviewer #3: No

 Response: the data was available in the form of tables.

4. Is the manuscript presented in an intelligible fashion and written in standard English?

Reviewer #1: No

Reviewer #2: Yes

Reviewer #3: No

Response: Thank you for your critical judgment, we have improved the language of the manuscript based on the comments given.

5. Review Comments to the Author

Response: Thank you very much for your concern.

 Reviewer #1: Major issues:

1. The Introduction and Methodology sections are clear, but the Discussion of key results is very weak. Results need to be discussed in a way that brings out the ‘big picture’ implications of the key findings.

Response: Thank you for the positive remark on our manuscript and we really appreciate your scientific judgment. The discussion part of the manuscript is modified and improved (please see the revised manuscript with track change document)

 Reference to literature from within and outside Ethiopia needs to be improved.

Response: We have added and refer different literature related to our title within and outside Ethiopia (please see the reference part from reference 37-49 on page 29)

2. The Results and Discussion section has numerous grammatical errors. The authors are advised to seek editing assistance from a native English speaker.

Response: Thank you for this comment and we addressed the issue of language editing. The manuscript is reviewed and edited by English language editor (professional). Hence, we believe that the language of the current version of the paper is significantly improved and consequently the quality of the paper too. (You can see revised manuscript with track changes document).

Minor issues:

1. The document requires line numbers in order to facilitate easier review process

Response: Fixed.

2. Avoid using words that already appear in the title as key words.

Response: Fixed.

3. Introduction section

• Do not start sentences with ‘And’

Response: improved

 Change tone ha-1 yr-1 to t ha-1 yr-1

Response: edited

• Farmers are mobilised through food-for-work programs, not through food & work programs.

Response: edited

Once the abbreviation for ‘Soil and Water Conservation’ has been defined in the Introduction, then use SWC throughout the rest of the document. There are instances where even both are used in the same sentence, avoid that.

Response: Edited

4. Materials & Methods section

• Scientific names of crops should be accompanied by the naming authorities

• Clarify how the selection of each kebele in the different watershed positions was done. Was it random or systematic?

Response: Thank you very much, we have explained the selection procedures (Please see page 6 line number 160-168)

• Replace ‘plot of land’ by ‘land’

Response:edited

• In the text, be consistent with the refencing style, either numbers or author name(s) followed by year. Check the journal requirement.

Response: We have improved the reference style based on PLOS ONE style

• The different levels of the sub titles should be clear.

Response: Edited

Results

• Tables are missing

Response: We have included the tables in the main body of the manuscript,

• Avoid putting table titles between paragraphs

Response: Dear reviewer, we thank you for these key comments and we agreed with all your ideas and the manuscript is already updated accordingly and please see revised manuscript with track changes document.

Reviewer #2: 

This paper examines the factors associated with adoption of sustainable soil and water conservation (SWC) practices in South Wollo, Ethiopia using primary survey data collected from 304 farm households. The data indicate that 64% of farmers adopted at least one of seven sustainable agricultural practices. The analysis finds that adoption is positively associated with family labor, education, and membership in local institutions. Factors found to correlate negatively with adoption are distance to market, engagement in off-farm activities, and the ratio of cultivable land to family size. While the article focuses on an important policy issue – how to encourage adoption of sustainable agricultural practices – it is not immediately evident how the results contribute to scholarship or are of interest to policymakers and practitioners outside Ethiopia.

 Response: Your comment is reasonable. Thank you. We made the revision according to your comments and please see revised manuscript with track changes document.

The author might therefore consider submitting the paper to a regional agricultural economics journal after careful revision. Below I provide some suggestions for how to improve the paper.

Response: We appreciate your positive reflection about the improvement of our study and thank you very much.

The Introduction includes useful background on agriculture and natural resource degradation in Ethiopia but leaves out some essential components, including (a) a brief review of relevant literature and the research gaps that merit investigation, (b) a statement of the research questions to be addressed or hypotheses to be tested, and (c) how the current work contributes to scholarship. In terms of contributions, the authors mention that our knowledge of SWC technology adoption is incomplete, because many studies consider a single SWC technology rather than multiple technologies. This is inaccurate. There are many studies, including for Ethiopia, that consider adoption of multiple technologies. The other contribution mentioned by the authors is the focus on South Wollo. What is it about South Wollo that makes it particularly interesting as a case study? And what do we learn from this study that has relevance regionally or globally? Those are important questions to address when highlighting the study’s contributions.

Response: We are grateful to the reviewer for his/her positive assessment of the manuscript. Following the reviewer comment, the manuscript has been amended. (please see revised manuscript with track changes document).

The manuscript appears to be technically sound, and the data do support the conclusions.

Response: Thank you very much for your positive comment.

However, the analyses have not been performed rigorously. The regression model is a logit model of the adoption of at least one of the seven SWC practices under study. A better approach is to model jointly the decisions to adopt different agricultural technologies, as is now common practice in the agricultural technology adoption literature (e.g., Kassie et al. 2013; Ward et al., 2018). These studies show that by analyzing the complementarity and substitutability of agricultural practices the drivers of adoption and overall uptake are better understood. It is recommended that the authors review the study of Teklewold et al. (2013) who modeled the adoption of sustainable agricultural practices by Ethiopian maize farmers as a joint decision and incorporated the extent of such adoption (number of practices adopted). The authors should consider using this modeling approach.

Response: We believed that this suggestion is valid and we appreciate your observation. However, We have given a detail justification for binary logistic regression analysis. (Please see page 8 line number27-35). That is: regression model, and its binary outcomes, helps the researcher to explore how each explanatory variable affects the probability of the occurrence of events [39]. This model helps to explore the degree and direction of the relationship between dependent and independent variables in the adoption of improved soil conservation technology at the household level [39]. The logistic regression model is an appropriate statistical tool to determine the influence of independent variables on dependent variables when the dependent variable has only two groups (dichotomous), e.g., adopters and non adopters, and the explanatory variables are continuous, categorical and dummy [6,10,16,32,38,39].It enabled to determine the impact of multiple independent variables on the dependent variable. Moreover, it is also widely applied in adoption decisions.

Two other suggestions for the empirical analysis are as follows: First, more information should be provided on how some variables were measured in the survey, to give readers a sense of their accuracy. For example, was farm size measured by farmer report, tape/compass, or GPS device? How were the variables for distance to market and farmland measured?

Response: To make it clear, we elaborate how the variables were measured page 7 line number from 179-182 and table 1. Farm size, distance to market and farmland were measured by farm report and tape/compass. Thank you very much. 

Second, the authors should include explanations for why the explanatory variables are included in the model, referring either to economic theory or previous empirical work.

Response: Thank you. We have included the explanations for the selection of variables.. It is based on previous empirical work. (please see page 10 line number from 267-272

Another possible way to increase the paper’s scholarly contribution would be to make better use of the qualitative data that were collected through focus group discussions (FGDs) and in-depth interviews. 

Response: Thank you for your suggestion. We have improved the manuscript ( please see page 13 and 14 line number 342-350)

The authors claim the study is mixed methods, but they only mention a few qualitative results. As described by Greene et al. (1989), there are five broad purposes of MM research (development, triangulation, complementarity, initiation, and expansion). The authors could consider which of these purposes is/are of most relevance to their study and use that to guide how they make use of the qualitative data. In terms of complementarity, it would be useful to present both summaries and illustrative quotes from the qualitative data to add richness to the quantitative findings.

Response: We have revised the manuscript based on this suggestion (please see revised manuscript with track changes document).

The Results section is written in a descriptive manner. This is acceptable, as long as a Discussion section is included, which is a requirement of PLOS One. The discussion is important for placing the results in context. In this section, the authors can return to the study hypotheses and discuss whether the findings are supportive of them. The discussion section should also refer to related studies and discuss how the findings of the present study agree or disagree with previous work and offer possible explanations. 

Response: Thank you very much for your scientific comments. We have discussed in detail the present findings with other findings within and outside Ethiopia (please see the result and discussion part in revised manuscript with track changes document).

Finally, it is important to highlight the implications of the study findings for policy and practice.

Response: I have indicated the implication in the discussion and conclusion part.

As a final recommendation, I suggest the authors carefully edit the manuscript for brevity (there is repetition in some areas) and grammar.

Response: Thank you for this comment and we addressed the issue of language editing. The manuscript is reviewed and edited by English language editor (professional). Hence, we believe that the language of the current version of the paper is significantly improved and consequently the quality of the paper too. (You can see revised manuscript with track changes document).

References:

Greene, J. C., Caracelli, V. J., and Graham, W. F. 1989 “Toward a Conceptual Framework for mixed method evaluation designs.” Educational Evaluation and Policy Analysis 11: 255-274.

Kassie, M., Jaleta, M., Shiferaw, B., Mmbando, F., and Mekuria, M.. 2013. “Adoption of Interrelated Sustainable Agricultural Practices in Smallholder Systems: Evidence from Rural Tanzania.” Technological Forecasting and Social Change 80(3): 525–540.

Teklewold, H., Kassie, M., and Shiferaw, B.. 2013. “Adoption of Multiple Sustainable Agricultural Practices in Rural Ethiopia.” Journal of Agricultural Economics 64(3): 597–623.

Ward, P.S., Bellb, A.R., Droppelmann, K., and Bentond, T.G.. 2018. “Early Adoption of Conservation Agriculture Practices: Understanding Partial Compliance in Programs with Multiple Adoption Decisions.” Land Use Policy 70(1): 27–37.

Response: Thank you very much for your concern. We have referred these articles.

Reviewer #3: 

Generally, such type of studies is timely because soil erosion is a significant environmental problem especially in countries like Ethiopia which depend on Agricultural activity which call for proper application of SWC activities. And identifying level of adoption and influencing factors are important. I thank you for sharing such local area-based study.

Response: We are grateful to the reviewer for his/her positive assessment of the manuscript.

However, the manuscript has number of limitations to meet publication and proper contribution of knowledge: - Poorly organized, and discussed (both abstract, background, result ….) and has a problem of flow of idea and coherence, some of them are redundant

Response: Following the reviewer comment, abstract, background and result section have been improved. Moreover, we tried to make the discussion precise and clear. (You can see revised manuscript with track changes document).

- Soil erosion and the practice is dynamic but the data and citation is too old (1980 & 1990’s…..)

Response: we have include latest reference ( Please see page 3 line number from 72-75)

- Incorporate the contribution of this study to SDG’s and Millennium Development goals of the country

Response: Thank you very much for your scientific comments and suggestions. We have included this issue. (Please see page 13 line number 326-341) 

.- It couldn’t consider the contribution of 60 days Mass mobilization conducted for two consecutive months (January and February) in the adoption of the practices. 

Response: Thank you. We have modified the manuscript ( Please see page 4 line number 91-96 and page 14 line number 337-356)

It is also difficult to differentiate the adopters from non-adopters

Response: We have indicated the explanation of adopters and non adopters on page 9 line number 257-262 in revised manuscript with track changes document

- You wrote “there was no adoption of SWC measures study in the study area, and it has been frequently hit by erosion hazards” as a gap, however, there are number of studies conducted in simlar areas (south Wollo, Amhara region…) which have the same social and physical setup … and it is not unique from other areas in erosion hazard, if it is you should explain numerically

Thus, I didn’t see its originality.

C Thank you. We have improved the manuscript.

In the Methodology part

- Justify the saying “even rains of moderate-intensity are enough to cause massive erosion” idea incorporate the annual soil loss rate and the erosivity and erodibility character of the RF and soil.

Response: Modified

- The sampling lacks clarity.

 Example

o The way you select the study area is not clear (it needs to justify how much it is vulnerable for erosion? by how much than other districts in the zone? “Long history of conservation” by how much than other woredas?

Response: We have improved the justification of sample selection procedures. (Please see page 6 line number 161-163)

- Usually, books and journals are used as literature not as data source. You stated that “secondary data were gathered from the department of Agriculture, books, and journals” . what type of data have you collected from books, journals and agriculture, where is the analysis of this data?

Response: Thank you for your positive comments. We have modified this issue in the manuscript.

- In statistical analysis :

o Which type of T-test have used for what purpose and which variable have you analyzed ? I debt the fitness of it for this study

Response: Modified. (Please see page 8 line number 218-221)

o How do you computed the dependency ratio, where is the result and data?

Response: dependency ratio: The ratio of total family number to the working age group (please see table 1)

- The map of study area is not to the standard. It lacks aesthetic value. The coloring of topography is proper (use Elevation 1 or 2 color ramp from ArcMap). The inset maps are too many which are irrelevant ( one “Ethiopia “ ) is enough , the legend, the grids( conducted), titling, scale writing and placement, and it is deformed. Thus, improve and meet the standard of study area map

Response: We have removed figure 1 because of the PLOS ONE recommendation

Result

- The model fitness and predictive capacity is very low (65% for non-adopters) why it is do you think the analysis result is reliable with this……………….; the multicollinearity?

Response: Before running the regression model, we have checked the multicollinearity among the explanatory variables. 

- I didn’t find any table in the manuscript (Table 1; 2;3 and 4) but it is sited in the discussion text. This made difficult to see the correctness of statistical values (analysis outputs). 

Response: Modified: We have put the tables in the main text of the manuscript.

Beside in the discussion part the result is discussed as “positively and significantly ..or …..” . without stating the value (p-value or wald statistics, and others…….) which shows how much the variable is significant than others. Generally, the statistical analysis result is not well interpreted and discussed with others study output.

Response: Your comment is reasonable. Thank you. We made the revision according to your comments and please see revised manuscript with track changes document.

 - The discussion is poor and shallow it seems a report than a research study. Use recent studies and discuss it the similarity and difference of your study from them and why, and how?

Response: Thank you very much for your comments. We have revised the manuscript based on this suggestion (please see revised manuscript with track changes document).

---

## [Decision Letter · Decision Letter 1]

21 Dec 2021

PONE-D-21-19431R1Farmers’ adoption of soil and water conservation practices: The case of Lege-Lafto Watershed, Dessie Zuria District, South Wollo, EthiopiaPLOS ONE

Dear Dr. Miheretu,

Thank you for submitting your manuscript to PLOS ONE. After careful consideration, we feel that it has merit but does not fully meet PLOS ONE’s publication criteria as it currently stands. Therefore, we invite you to submit a revised version of the manuscript that addresses the points raised during the review process.

 Progress has been made, but there are some revisions still needed. In your revision, take into account and address that the reviewers have take the time to indicate what needs to be done, and provided some important feedback. Please pay close attention to reviewer #2, there are still some issues to be addressed fully. The map as indicate by reviewer #3 is also needed, a google earth engine type map to better understand the location of the study will address this.

We look forward to receiving your revised manuscript.

Kind regards,

Sieglinde S. Snapp

Academic Editor

PLOS ONE

Journal Requirements:

Reviewers' comments:

Reviewer's Responses to Questions

**Comments to the Author**

1. If the authors have adequately addressed your comments raised in a previous round of review and you feel that this manuscript is now acceptable for publication, you may indicate that here to bypass the “Comments to the Author” section, enter your conflict of interest statement in the “Confidential to Editor” section, and submit your "Accept" recommendation.

Reviewer #2: (No Response)

Reviewer #3: All comments have been addressed

2. Is the manuscript technically sound, and do the data support the conclusions?

Reviewer #2: Yes

Reviewer #3: Partly

3. Has the statistical analysis been performed appropriately and rigorously? 

Reviewer #2: No

Reviewer #3: Yes

4. Have the authors made all data underlying the findings in their manuscript fully available?

Reviewer #2: Yes

Reviewer #3: Yes

5. Is the manuscript presented in an intelligible fashion and written in standard English?

Reviewer #2: No

Reviewer #3: Yes

6. Review Comments to the Author

Reviewer #2: The study authors have made some improvements to the manuscript. A few of my earlier comments were addressed, but several comments were not adequately addressed as described below.

• The revised version has been professionally edited and this is much appreciated. However, some grammatical errors remain. Further professional editing may be needed.

• The Introduction was slightly revised, but the authors did not address my suggestions to (a) explicitly state the research gaps that the current study addresses and (b) explicitly state the research questions to be studied or hypotheses to be tested. And I’m still not clear on what it is about South Wollo that makes it particularly interesting as a case study.

• I maintain that the analyses have not been performed rigorously. A better approach (vs. the current single equation logit model) is to model jointly the decisions to adopt different agricultural technologies, as is now common practice in the agricultural technology adoption literature (e.g., Kassie et al. 2013; Ward et al., 2018). These studies show that by analyzing the complementarity and substitutability of agricultural practices the drivers of adoption and overall uptake are better understood. It is recommended that the authors review the study of Teklewold et al. (2013) who modeled the adoption of sustainable agricultural practices by Ethiopian maize farmers as a joint decision and incorporated the extent of such adoption (number of practices adopted). The authors should consider using this modeling approach.

• The authors have addressed my comment to include explanations for why the explanatory variables are included in the model, by citing previous empirical work. Thank you.

• I appreciate that the authors have tried to make better use of the qualitative data that were collected through focus group discussions (FGDs) and in-depth interviews. However, I see only one quote was added. And the FGD and KII results are only mentioned in one other part of the paper (lines 367-372). The authors should add several more quotes and summarize some of the other qualitative results at appropriate points in the paper to complement the quantitative results. That would then support the authors claim that the study is mixed methods.

• The authors have improved the Results section by weaving in a few results of related studies and discussing how the findings of the present study agree or disagree with previous work and offering plausible explanations. Thank you.

• The authors have not adequately highlighted the implications of the study findings for policy and practice. I suggest this be done in the next round of revisions.

Reviewer #3: Authors, thank you, most of the issue were addresses but still some issue needs clarification and correction :- to accept the manuscript for publication the following issues should be addressed.

1. Line 126....."The mean annual rainfall of the area is 1365.7 mm with a maximum of 2074.4mm in 2018 and a

127 minimum of 902.4mm in 2007. " what does it mean why it is data for two (2007 & 2018) ?

2. Line 154 ...... "the survey population and the number of sample households was decided by using the [33] formula." would be clear if you wrote the name of the formula and followed by refere.. (33)

3. Line 157. from the total (375) questioners, 53 samples were not properly respond thus discarded from the analysis. this number nearly 14% of the total sample. How do you maintain representativeness of the data and how do you managed it.

4. as the study is geographic, the map should be there. deal with the journal editor and incorporate the map of the study watershed . It is mandatory having a map to locate and show the position of the study area for any body.

5. still the language needs improvement.

7. PLOS authors have the option to publish the peer review history of their article (what does this mean?). If published, this will include your full peer review and any attached files.

Reviewer #2: No

Reviewer #3: No

---

## [Author Response · Author response to Decision Letter 1]

15 Feb 2022

Dear Sieglinde S. Snapp

Academic Editor

PLOS ONE

Subject: Ref.: No. PONE-D-21-19431R1

We are writing this with reference to the revised manuscript entitled Farmers’ adoption of soil and water conservation practices: The case of Lege-Lafto Watershed, Dessie Zuria District, South Wollo, Ethiopia, which has be submitted for publication in PLOS ONE. 

We would like to express our sincere gratitude to the reviewers for their valuable comments and suggestions. We have revised the manuscript by taking into account the comments and suggestions given by reviewers (Reviewers #2 and #3). We are now re-submitting the revised version for your kind reconsideration for publication. We sincerely hope that we have sufficiently addressed the suggested comments, and have prepared the manuscript in accordance with the journal’s style. The details of how the comments and suggestions as addressed point by point are given below. 

Looking forward to hear your positive response to publish our manuscript in your Journal, 

=====

Rebuttal letter

Response to the Journal Requirements

Journal Requirements:

Response: The reference list is complete and correct.

Reviewers' comments:

Reviewer's Responses to Questions

Comments to the Author

1. If the authors have adequately addressed your comments raised in a previous round of review and you feel that this manuscript is now acceptable for publication, you may indicate that here to bypass the “Comments to the Author” section, enter your conflict of interest statement in the “Confidential to Editor” section, and submit your "Accept" recommendation.

Reviewer #2: (No Response)

Reviewer #3: All comments have been addressed

Response: Thank you very much

2. Is the manuscript technically sound, and do the data support the conclusions?

Reviewer #2: Yes

Reviewer #3: Partly

Response: Thank you very much

3. Has the statistical analysis been performed appropriately and rigorously?

Reviewer #2: No

Reviewer #3: Yes

Response: Thank you very much for your comments. We have done the statistical analysis properly.

4. Have the authors made all data underlying the findings in their manuscript fully available?

Reviewer #2: Yes

Reviewer #3: Yes

Response: Thank you very much

5. Is the manuscript presented in an intelligible fashion and written in standard English?

Reviewer #2: No

Reviewer #3: Yes

Response: Edited

6. Review Comments to the Author

Response: Thank you very much

Reviewer #2: The study authors have made some improvements to the manuscript. A few of my earlier comments were addressed, but several comments were not adequately addressed as described below.

• The revised version has been professionally edited and this is much appreciated. 

Response: Thank you

However, some grammatical errors remain. Further professional editing may be needed.

Response: The recent version of the revised manuscript also professionally edited.

• The Introduction was slightly revised, but the authors did not address my suggestions to (a) explicitly state the research gaps that the current study addresses 

Response: Thank you again for your comments. We have indicated the research gap (please see page 4 of the revised manuscript with track changes word document)

and (b) explicitly state the research questions to be studied or hypotheses to be tested. 

Response: Fixed. Please see line number 111-112 on page 4 in revised manuscript with track changes word document)

And I’m still not clear on what it is about South Wollo that makes it particularly interesting as a case study.

Response: We have clearly described again the situation of South Wollo. (Please see line number 102-111 on page 4 in revised manuscript with track changes word document)

• I maintain that the analyses have not been performed rigorously. A better approach (vs. the current single equation logit model) is to model jointly the decisions to adopt different agricultural technologies, as is now common practice in the agricultural technology adoption literature (e.g., Kassie et al. 2013; Ward et al., 2018). These studies show that by analyzing the complementarity and substitutability of agricultural practices the drivers of adoption and overall uptake are better understood. It is recommended that the authors review the study of Teklewold et al. (2013) who modeled the adoption of sustainable agricultural practices by Ethiopian maize farmers as a joint decision and incorporated the extent of such adoption (number of practices adopted). The authors should consider using this modeling approach.

Response: Thank you again for your comments. We believed that this suggestion is valid and we appreciate your observation again. However, we have given a detail justification for binary logistic regression analysis in the manuscript. Moreover, we have used the references that are widely applied binary logistic regression in adoption decisions.

• The authors have addressed my comment to include explanations for why the explanatory variables are included in the model, by citing previous empirical work. Thank you.

Response : Thank you very much

• I appreciate that the authors have tried to make better use of the qualitative data that were collected through focus group discussions (FGDs) and in-depth interviews. However, I see only one quote was added. And the FGD and KII results are only mentioned in one other part of the paper (lines 367-372). The authors should add several more quotes and summarize some of the other qualitative results at appropriate points in the paper to complement the quantitative results. That would then support the authors claim that the study is mixed methods.

Response: Thank you for your constructive comments and suggestions. We have added other explanations in the manuscript. (Please see line number 306-311 on page 12 and 13, and page 14 line number 359-365 in revised manuscript with track changes word document)

• The authors have improved the Results section by weaving in a few results of related studies and discussing how the findings of the present study agree or disagree with previous work and offering plausible explanations. Thank you.

Response : Thank you very much

• The authors have not adequately highlighted the implications of the study findings for policy and practice. I suggest this be done in the next round of revisions.

Response: Thank you very much for your comments: We have included the implication of the study in the revised manuscript. (Please see line number 538-555 on page 23 in revised manuscript with track changes word document)



Reviewer #3: Authors, thank you, most of the issue were addresses but still some issue needs clarification and correction :- to accept the manuscript for publication the following issues should be addressed.

Response: Thank you very much for your positive response

1. Line 126....."The mean annual rainfall of the area is 1365.7 mm with a maximum of 2074.4mm in 2018 and a

127 minimum of 902.4mm in 2007. " what does it mean why it is data for two (2007 & 2018) ?

Response: rewrite it. (Please see line number 135-139 on page 5 and 6 in revised manuscript with track changes word document)

2. Line 154 ...... "the survey population and the number of sample households was decided by using the [33] formula." would be clear if you wrote the name of the formula and followed by refere.. (33)

Response: Fixed.

3. Line 157. from the total (375) questioners, 53 samples were not properly respond thus discarded from the analysis. this number nearly 14% of the total sample. How do you maintain representativeness of the data and how do you managed it.

Response: In fact, the method that we employed for sample size determination is a bit firm. As far as other methods of sample size determination are concerned, we have sampled more farmers and the data could be little affected by the discarded samples in terms of representativeness.

4. as the study is geographic, the map should be there. deal with the journal editor and incorporate the map of the study watershed . It is mandatory having a map to locate and show the position of the study area for any body.

Response: Thank you very much for your critical observation. We have included the revised study area map in the manuscript.

5. still the language needs improvement.

Response: The recent version of the revised manuscript also professionally edited.

7. PLOS authors have the option to publish the peer review history of their article (what does this mean?). If published, this will include your full peer review and any attached files.

Do you want your identity to be public for this peer review? For information about this choice, including consent withdrawal, please see our Privacy Policy.

Reviewer #2: No

Reviewer #3: No

---

## [Editor Report · Decision Letter 2]

23 Feb 2022

Farmers’ adoption of soil and water conservation practices: The case of Lege-Lafto Watershed, Dessie Zuria District, South Wollo, Ethiopia

PONE-D-21-19431R2

Dear Dr. Miheretu,

We’re pleased to inform you that your manuscript has been judged scientifically suitable for publication and will be formally accepted for publication once it meets all outstanding technical requirements.

One remaining issue: it is important to provide confidentiality for the farmer's whose name is given with the quote by removing his name, instead you can just describe him as a farmer of X age, as done for other quotes.

Kind regards,

Sieglinde S. Snapp

Academic Editor

PLOS ONE

---

## [Editor Report · Acceptance letter]

8 Mar 2022

PONE-D-21-19431R2 

Farmers’ adoption of soil and water conservation practices: The case of Lege-Lafto Watershed, Dessie Zuria District, South Wollo, Ethiopia 

Dear Dr. Miheretu:

I'm pleased to inform you that your manuscript has been deemed suitable for publication in PLOS ONE. Congratulations! Your manuscript is now with our production department. 

Kind regards, 

on behalf of

Dr. Sieglinde S. Snapp 

Academic Editor

PLOS ONE